# Connexins and Pannexins: Important Players in Neurodevelopment, Neurological Diseases, and Potential Therapeutics

**DOI:** 10.3390/biomedicines10092237

**Published:** 2022-09-09

**Authors:** Daniela Baracaldo-Santamaría, María Gabriela Corrales-Hernández, Maria Camila Ortiz-Vergara, Valeria Cormane-Alfaro, Ricardo-Miguel Luque-Bernal, Carlos-Alberto Calderon-Ospina, Juan-Fernando Cediel-Becerra

**Affiliations:** 1Pharmacology Unit, Department of Biomedical Sciences, School of Medicine and Health Sciences, Universidad del Rosario, Bogotá 111221, Colombia; 2Anatomy and Embriology Units, Department of Biomedical Sciences, School of Medicine and Health Sciences, Universidad del Rosario, Bogotá 111221, Colombia; 3GENIUROS Research Group, Center for Research in Genetics and Genomics (CIGGUR), School of Medicine and Health Sciences, Universidad del Rosario, Bogotá 111221, Colombia; 4Histology and Embryology Unit, Department of Biomedical Sciences, School of Medicine and Health Sciences, Universidad del Rosario, Bogotá 111221, Colombia

**Keywords:** connexins, gap junctions, neurodegenerative diseases, embryonic development

## Abstract

Cell-to-cell communication is essential for proper embryonic development and its dysfunction may lead to disease. Recent research has drawn attention to a new group of molecules called connexins (Cxs) and pannexins (Panxs). Cxs have been described for more than forty years as pivotal regulators of embryogenesis; however, the exact mechanism by which they provide this regulation has not been clearly elucidated. Consequently, Cxs and Panxs have been linked to congenital neurodegenerative diseases such as Charcot-Marie-Tooth disease and, more recently, chronic hemichannel opening has been associated with adult neurodegenerative diseases (e.g., Alzheimer’s disease). Cell-to-cell communication via gap junctions formed by hexameric assemblies of Cxs, known as connexons, is believed to be a crucial component in developmental regulation. As for Panxs, despite being topologically similar to Cxs, they predominantly seem to form channels connecting the cytoplasm to the extracellular space and, despite recent research into Panx1 (Pannexin 1) expression in different regions of the brain during the embryonic phase, it has been studied to a lesser degree. When it comes to the nervous system, Cxs and Panxs play an important role in early stages of neuronal development with a wide span of action ranging from cellular migration during early stages to neuronal differentiation and system circuitry formation. In this review, we describe the most recent available evidence regarding the molecular and structural aspects of Cx and Panx channels, their role in neurodevelopment, congenital and adult neurological diseases, and finally propose how pharmacological modulation of these channels could modify the pathogenesis of some diseases.

## 1. Introduction

During embryonic development, organs are formed by interactions between cells and tissues. Usually, one group of cells (the inducing tissue) influences the development of another set of cells (the responding tissue) in a process called induction [1]. Cell-to-cell communication is essential for induction, for the ability of a cell to respond, and to allow dialogue between inducing and responding cells [2]. Therefore, communication between cells is critical for embryonic development, and the main mechanisms that allow this communication are juxtracrine and paracrine signaling [3]. Juxtracrine signaling can occur in three principal ways: the first occurs when a protein on the surface of one cell interacts with a receptor on the other cell, the second involves ligands that are secreted by one cell to the extracellular matrix where they interact with the receptors on the other cell, and the third is the direct transmission of signals via gap junctions from one cell to another [4]. The latter is especially important in cells that are tightly connected, such as the epithelial cells or the neural tube cells [5]. It is here where gap junctions acquire a pivotal role in embryogenesis, helping to regulate cellular behaviors such as proliferation, migration, and specification by allowing intracellular communication between cells.

Gap junctions allow intracellular exchange by enabling the passage of small molecules less than 1.8 kDa in size, such as ions, small peptides, second messengers, and metabolites [6]. They are formed by the pairing of two hemichannels called connexons, said hemichannels are formed by hexameric assemblies of connexins (Cxs) that ultimately function to physically connect adjacent cells [6]. Cxs are named according to their molecular mass: in this way, a connexin (Cx) that has a molecular mass of 43 kDa is called Cx43. They express in a spatio-temporal specific pattern, meaning that certain Cx isoforms can only be found in a specific tissue at a specific time during development [7]. 

Furthermore, connexons can be formed by different isoforms of Cxs, or similarly, gap junctions can be assembled by different connexons, forming heteromeric connexons or heterotypic gap junctions, respectively [8]. Notably, the specific Cx isoform composition of a connexon or a gap junction has functional consequences: it can determine some properties such as pore size, voltage-dependent gating and permeability [9]. There are 21 identified Cxs in rodents and 20 in humans that share sequence homology (Figure 1), and, of these, Cx43 is the most abundant and studied [10]. Aside from allowing small molecules to pass between cells, gap junctions also have channel-independent functions, such as interaction with the cytoskeleton or cell-cell adhesion. The latter is important during the development of the central nervous system (CNS) because it allows the coupling of different cells to achieve a coordinated electrical transmission [11]. Furthermore, prior to pairing with another connexon and forming gap junctions, connexons can also secrete molecules such as potassium, glutamate, and adenosine triphosphate (ATP) into the extracellular space; thus, in the CNS they can modulate neuronal activity and influence different signaling pathways [12].

For their part, Pannexins (Panxs) are structurally similar to Cxs; however, they do not share homology in their sequence (Figure 2) [13]. Three members are described in this family—Panx1, Panx2, and Panx3—that, similarly to Cxs, can oligomerize to form higher order structures called pannexons. Panx1, in contrast with Cxs, has recently been discovered to oligomerize into heptamers [14], whereas Panx2 oligomerizes into octamers and Panx3′s stoichiometry is unknown. Even though they are structurally similar to Cxs (Figure 3), Panxs are not believed to be able to form gap junction channels, although this is still under debate [15,16,17]. Structurally, Panxs have 4 conserved cysteines, two in each extracellular loop, that in addition to the glycosylation of the first extracellular loop in Panx2 and the second in Panx1 make them different from Cxs [18]. Pannexons allow efflux of ATP and calcium to the extracellular space [19,20], where these molecules serve as signals for diverse physiological functions (e.g., ATP helps the propagation of calcium waves between astrocytes, can act as a neurotransmitter, and can induce apoptosis) [21]. In addition, ATP released by pannexons binds to purinergic receptors (P2X and P2Y) that can induce more ATP release by these channels, creating a positive feedback loop and an ATP-induced ATP release mechanism which in some conditions can be toxic [22]. 

Both Cxs and Panxs have been associated with essential functions during development and in the pathogenesis of many neurological diseases. They have been found to be expressed in the embryoblast [23], in subplate neurons (one of the earliest born neurons responsible for the regulation of early electrical activity) [24], during the early human development of the spinal cord, and in the three developing meningeal layers [25]. It has also been shown that they can modulate gene expression in neighboring cells [26], release neuroactive compounds (ATP, glutamate), modulate synaptic transmission, synaptic plasticity and spatial learning, maintain the integrity of the blood-brain-barrier (BBB), and regulate neural crest cell migration [5]. The participation of Cxs and Panxs in the formation of the peripheral nervous system (PNS) and CNS is evidenced when genetic variations in the genes that code for these channels result in congenital diseases such as Charcot-Marie-Tooth disease, oculodentodigital dysplasia, and Pelizaeus-Merzbacher-like disease 1. Cxs and Panxs are also involved in the pathogenesis of adult neurological diseases, namely neurodegenerative diseases, such as Alzheimer’s and Parkinson’s disease, multiple sclerosis, peripheral neuropathies, and most recently in ischemia [27,28]. 

Even though it is known that Cxs and Panxs are fundamental for the adequate function of the CNS and PNS, the exact mechanisms by which they mediate this are largely unknown. The topic has been widely studied, but new research has emerged considering these proteins as potential therapeutic targets [29], making a review necessary to characterize how pharmacological modulation could influence the pathogenesis of frequent neurological diseases that currently have no treatment.

In addition, it is important to review how these channels influence the development of cells of the CNS and PNS, since some of the mechanisms used during development are retained and adapted to mediate changes in the mature brain. For example, the potential for neurogenesis is retained in the hippocampus [30] and could be essential after ischemic injury, while the migration and differentiation of oligodendrocyte progenitor cells (OPCs) to areas of demyelination has an important impact in multiple sclerosis [31]. Consequently, the aim of this review is to describe the most recent available evidence regarding the molecular and structural aspects of Cx and Panx channels, their role in neurodevelopment, congenital and adult neurological diseases, and finally to describe the available studies that have evaluated the pharmacological modulation of these channels in various neurological diseases. 

## 2. Structure of Cxs and Panxs

In order to understand the role of these important proteins in the development and function of the nervous system, it is first important to understand the structure, gating kinetics, inhibition and known pharmacological agents of Cxs and Panxs. In this section, we will review the structure and function of Cx43 and Panx1, given that they are the two channel-forming proteins most frequently associated with disease and the most widely studied.

Cxs are structurally similar: they have 4 hydrophobic transmembrane domains (M1-M4), two extracellular loops, one cytoplasmic loop, and free cytosolic N- and C-termini. The extracellular loops work to connect to the Cxs of neighboring cells, while the N-terminal and the C-terminal have been associated with small molecule selectivity [32]. Most of the information known today about the molecular biology of Cxs has been derived from studies on Cx43. However, recently many gap junctions and hemichannels have been structurally characterized at high resolution using X-ray crystallography and cryo-EM, presenting insights into the shared structural similarities among Cxs. 

For instance, comparative analysis between Cx26 [33] and Cx46/50 gap junctions [34], as well as the structure of Cx31.3 hemichannels [35] showed that the N-terminal domain appears to be involved in the gating mechanisms of gap junctions and hemichannels, adopting several conformations in different Cxs [36]. Disease-related mutations seem to be located in the extracellular loops, the pore region and the gating region. It is also known that post-translational modifications at the cytoplasmic C-terminal such as phosphorylation can directly regulate gap junction opening [37]. For its part, despite a lack of sequence homology, the cryo-EM structure of human Panx1 at 2.8 Å resolution has shown that it has a similar quaternary structure to Cxs [13].

### 2.1. Cx43

Cx 43, similar to other Cxs, has 4 transmembrane domains, cytosolic N- and C-termini, two extracellular loops and one intracellular loop. On the other hand, in contrast with Panxs, Cxs do not exhibit glycosylation of the extracellular loops and can therefore form gap junctions. Cx43-based channels can be regulated by many factors, such as voltage, pH, divalent ions, reactive oxygen species, phosphorylation, and elevated intracellular calcium levels. The gating kinetics of Cxs is a matter of ongoing research, but recent studies have provided some insights. It appears that the N-terminal helix plays a critical role in channel gating and charge selectivity. At a pH of 8.0, the six N-terminal helixes in each connexon adopt a gate-covering conformation representing the closed state of the channel. 

At a pH of 6.9, the open state is favored: the transition between the closed and open state involves structural changes (in the secondary structure) in the first transmembrane domain, movement of the second to the fourth transmembrane domain and structural stabilization of the cytoplasmic loop [36,38]. The pharmacological inhibition of hemichannels formed by Cx43 have been tested in cerebral ischemia and in myocardial ischemia models, in which a large number of hemichannels are usually opened. These models principally use gap19, a potent and selective blocker of Cx43 hemichannels. Gap19 is a peptide that inhibits the hemichannel opening by preventing interaction between the C-terminus and the intracellular loop, without interfering with gap junction communication between astrocytes. This specific inhibition has been shown to be neuroprotective after cerebral ischemia [39] and even to exert anticonvulsant effects [40] in murine models. 

### 2.2. Panx1

Panx1, similar to its counterparts, has 4 transmembrane domains (M1-M4), cytosolic N- and C-termini, two extracellular loops and one intracellular loop. Until recently, Panx1 was thought to oligomerize as hexamers, but new cryo-EM structures have revealed the channels to be heptameric [41,42,43]. Panx1 can be activated by mechanical stretching, extracellular concentrations of K^+^, intracellular concentrations of Ca^2+^, opening of the P2X7 channel, tyrosine phosphorylation at the intracellular loop, or by membrane depolarization [14,44,45]. 

Some extracellular loops of Panx1 may exhibit glycosylation, which interestingly may be the reason for their lack of ability to form gap junctions, as demonstrated by glycosylation-deficient Panx channels. As for gating kinetics, a recent study demonstrated that the C-terminal tail acts as a gate, blocking the intracellular entry until its cleavage by caspase 3 or 7 opens the channel, suggesting a caspase-dependent gating mechanism [14]. 

Cleavage of the C-terminal tail results in channel activation and release of ATP, a mechanism that was initially thought to occur only during apoptosis. Nonetheless, ionic conductance has been documented in the absence of caspase treatment. The hypothesis for this caspase-independent gating is attributed to the linker connecting the N-terminal helix to the M1 [14,46]. Now, some compounds such as carbenoxolone have been shown to inhibit the activity of Panx1; however, the exact binding sites were not known until recently. It is now believed to bind to the extracellular entrance that contains tryptophan-74. 

## 3. General Role of Cxs and Panxs in the Nervous System

Gap junctions are required for intracellular communication, but they also perform channel-independent tasks such as interactions with cytoskeleton proteins and cell-cell adhesion. Apart from their crucial activities, the discovery that Cxs are more than channel-forming proteins raised interest, as did their involvement in embryology. Many studies began documenting the expression of Cx genes during embryonic development as protein or messenger RNA (mRNA). Early studies evidenced the expression of Cx31, Cx43, and Cx45 in the mouse embryoblast, as well as gap junction intracellular communication through tracer injection [23,47]. Later on, the expression of Cx43, Cx45 and Cx40 as proteins was also documented in human embryonic stem cells [48,49]. Thereafter, the role of Cxs in neurodevelopment began to be investigated. For instance, there is evidence that during the development of the cortex, human subplate neurons, which are one of the earliest-born neurons responsible for the regulation of early electrical activity between the 17th and the 23rd gestational week [24], express Cx26, Cx32 and Cx36, as detected by protein chain reaction (PCR) or immunohistochemistry [50].

Cxs may also be involved in differentiation, proliferation, and apoptosis. Exactly how Cxs are involved in these processes is not entirely elucidated; however, there are two possible mechanisms that have been described. The first proposes that the intracellular diffusion of IP_3_ (Inositol Triphosphate) and Ca^2+^, allowed by gap junctions, modulate gene expression in neighboring cells in a synchronized manner, via calcineurin/NFAT (Nuclear factor of activated T-cells) signaling, a signaling pathway implicated in the development of cardiac and skeletal muscle, and the nervous system [23,26,51]. Cxs could also promote the activation of Ca^2+^signaling through the release of ATP by hemichannels into the interstitial space. ATP activates purinergic receptors on the surface of the embryonic stem cell, which in turn leads to the mobilization of intracellular Ca^2+^, activating various pathways such as calcineurin/NFAT-signaling [12,23]. Table 1 summarizes the role of each Cx during development and in the adult CNS and PNS.

While Cxs have shown relevant roles during the development of the CNS, they also have important functions in the adult brain. The main purpose of Cx-based channels is to help to maintain brain homeostasis by allowing communication within the neuro-glia-vascular unit [52,53,54]. Cx30 and Cx43 are widely expressed in astrocytic endfoot processes that contact cerebral vasculature, forming part of the blood-brain barrier (BBB). Consequently, recent studies show that Cx43 is essential for maintaining BBB function [55].

**Table 1 biomedicines-10-02237-t001:** Functions of the relevant Cxs expressed during development, in the adult brain, and related neurological diseases.

Cx	Gene	Relevant Function during Development	Relevant Function in Adult CNS or PNS	Related Diseases and Functional Alteration	Reference
Cx43	Gap junction protein alpha 1	◦ Specification of pre-migratory cranial neural crest cells ◦ Regulation of neural cell crest migration by regulating the transcription of N-cadherin ◦ Coupling of mesenchymal stem cells ◦ Morphological maturation of astrocytes ◦ Regulation of astrocyte migration	◦ Contributes to neural homeostasis, hippocampal plasticity, and spatial cognition in mice ◦ Principal Cx in astrocytic network ◦ Contributes to formation and maintenance of CNS myelin by helping to regulate oligodendrocyte nutrient homeostasis ◦ Maintains NPCs in a non-differentiated and proliferative state in the SVZ	◦ Neurodegenerative diseases ◦ Oculodentodigital dysplasia (≈70 causative mutations) ◦ Increased expression in MS	[5,53,56,57]
CX37	Gap junction protein alpha 4	-	◦ Expressed in vascular endothelial cells that form part of the BBB	-	[52]
CX40	Gap junction protein alpha 5	◦ Coupling of mesenchymal stem cells	◦ Expressed in vascular endothelial cells that form part of the BBB	-	[52,58]
CX32	Gap junction protein beta 1	◦ Contributes to spontaneous membrane depolarizations in subplate neurons in the developing cortex	◦ Homeostasis of myelinated axons	◦ Charcot-Marie-Tooth disease	[50,59]
CX26	Gap junction protein beta 2	◦ Coupling of excitatory neurons in the developing neocortex (rodent study)	◦ Forms heterotypic GJs between astrocytes and oligodendrocytes	◦ Sensorineural deafness, hearing loss and hyperkeratosis, missense mutation in *GJB2* gene results in inability to form gap junctions	[60,61]
CX30	Gap junction protein beta 6	◦ Regulates the extension and ramification of astroglia processes	◦ Inter-astrocyte coupling ◦ Neural homeostasis, hippocampal plasticity, and spatial cognition	Upregulation in neurodegenerative diseases	[62,63]
CX45	Gap junction protein gamma 1	◦ Coupling of mesenchymal stem cells	◦ Inter-astrocytic gap junctions	-	[58]
CX47 CX46.6	Gap junction protein gamma 2	-	◦ Contributes to formation and maintenance of CNS myelin by helping to regulate oligodendrocyte nutrient homeostasis	Pelizaeus-Merzbacher-like disease 1	[64,65]
CX36	Gap junction protein delta 2	◦ Regulates neuronal differentiation from neural progenitor cells ◦ Contributes to spontaneous membrane depolarizations in subplate neurons in the developing cortex	◦ Synaptic plasticity modulation in the hippocampus, memory impairment, and changes in motor performance in Cx36-deficient mice	-	[66,67]

Regarding Panxs, Panx1 is the most widely expressed across different tissues: in the CNS it is more prominent in the cerebellum, hippocampus, olfactory bulb, and thalamus, while it is also present in the sensory ganglia of the PNS. Panx2 is principally detected in the cerebellum, spinal cord, and brain stem. Panx 3 is not expressed in the nervous system. The expression and function of Panxs during early development of the CNS is less understood. A strong amount of evidence documents a link between Panx1 and P2X7 receptors, which are essential for neural and mesenchymal stem cell maintenance and differentiation [68,69,70]. Thus, Panx1 has been associated with proliferation of neural stem and progenitor cells through the release of ATP, both in vitro and in vivo [30,71]. Furthermore, the expression of Panx1, Panx2 and Panx3 mRNA have also been found in human pluripotent stem cells (e.g., human embryonic stem cells) [72]. Many diseases have been associated with Panx, mainly epilepsy, chronic pain and stroke, which will be reviewed later [73,74]. Table 2 summarizes the role of each Panx during development and in the adult CNS and PNS.

### 3.1. Role of Cxs and Panxs in Cells of the CNS

For this review, we will refer to the multipotent cells that give rise to the majority of the glial and neuronal cell types that populate the CNS as neural progenitor cells (NPCs), and we will use NPCs interchangeably with radial glia [77]. 

#### 3.1.1. Neurons 

Cxs and Panxs are involved in the generation and development of new neurons in the embryonic, postnatal and adult brain. The majority of neurons develop during the embryonic period from multipotent cells collectively known as NPCs located in the embryonic ventricular zone. There are many ways in which Cx expression can impact embryonic neurogenesis. First, they play a crucial role in neuronal differentiation, for example Hartfield et al., demonstrated that neuronal differentiation is positively influenced by Cx36 in a neurosphere model [67]. Cx36 knockdown resulted in significantly reduced neuronal differentiation, while increased expression of Cx36 resulted in increased neurogenesis [67]. The authors hypothesized that increased Cx36 expression provides an enhanced cell-cell contact between NPCs, which promoted neuronal differentiation. However, this increase/decrease in neurogenesis with over expression or knockout of Cx36 was only seen in cells of 7 days or less of age, while cells older than this had already become committed and did not respond to changes in Cx36 expression [67]. 

Furthermore, Cx43 has been implicated in neuronal migration. Qi et al., documented that Cx43 knockout impaired neuronal migration and formation of a laminar structured cortex during embryonic brain development. Cx43 is not expressed in the mature neuron, in fact Cx43 gradual decrease is necessary for neuronal differentiation: the authors further encountered that said Cx43 gradual decrease was due to upregulation of cyclin-dependent kinase 5, which phosphorylates Cx43 at Ser279 and Ser282, promoting its degradation [78]. 

In the postnatal brain, Swayne et al., demonstrated that Panx2 is dynamically expressed during hippocampal neurogenesis, and they found Panx2 knockdown significantly accelerated the rate of neuronal differentiation suggesting a modulatory role in neurogenesis, regulating the timing to differentiation once exposed to neurogenic stimuli [79]. Furthermore, Wang et al., demonstrated that mice with Cx43 knockout in excitatory neurons during the early postnatal stages had impaired short-term motor learning capacity [80].

Neurogenesis in the adult brain has also been documented in specific areas. Newborn neurons in the adult originate from NPCs located in the subgranular zone of the dentate gyrus (SGZ) and in the subventricular zone (SVZ) of the lateral ventricle. These NPCs express Cx26, Cx43, Cx45 and Panx1 in vitro, and have been demonstrated to have hemichannel activity and form functional gap junctions [81]. Cx43 is highly expressed in NPCs and serves a number of functions: firstly, it maintains NPCs in a non-differentiated and proliferative state, secondly it promotes neuroblast migration, and thirdly it maintains the cellular architecture of the SVZ niche [82,83].

#### 3.1.2. Astrocytes

Astrocytes, as well as oligodendrocytes and neurons, derive from NPCs located in the ventricular zone of the developing CNS, and will develop after neurons, having a peak in late prenatal to early post-natal stages (embryonic day 18 to postnatal day 7). Cxs play an important role during both the development (migration, differentiation, and morphological maturation) and in the normal function of the mature cell (contribution to the BBB, maintenance of ion balance, enhancement of myelination by oligodendrocytes, synaptic regulation, and formation of the astrocytic network). Both Cx43 and Cx30 have been implicated in the morphological maturation of astrocytes. Cultures of mouse astrocytes with Cx43 knockdown showed upregulation of a group of cytoskeletal proteins including actin, tropomyosin, microtubule-associated protein RP/EB1, transgelin, and glial fibrillary acidic protein (GFAP) [84]. These proteins play important roles in diverse cellular functions such as attachment, motility, and shape formation. The authors concluded that downregulation of Cx43 expression led to changes in morphology, migratory activity, and cell adhesion of astrocytes [84]. 

Ghézali et al., later demonstrated that despite the late postnatal expression of Cx30, it also regulates the extension and ramification of astroglia processes, and that its expression sets the polarization of astroglia protrusions in migrating astrocytes via modulation of laminin/β1 integrin cascade, meaning Cx30 plays a crucial role in the postnatal morpho-functional maturation of astrocytes [85]. Cx43 and Cx30 are also widely expressed in the mature astrocyte, although they have different expression patterns. Cx43 is expressed from the embryonic to the adult stage and is highly expressed in the hippocampus and cerebellum. Cx30 expression, on the other hand, begins after the second postnatal week, can be regulated by neuronal activity [85] and is highly expressed in the thalamic and hypothalamic areas. 

Cx43 and Cx30 play crucial roles in the function of the mature cell: they allow the formation of highly interconnected astrocytes that give rise to the astroglial network, responsible for providing energy supply to neurons and supporting synaptic activity. For instance, some of the functions carried out by the astroglial network that are dependent on Cxs and gap junctions are the diffusion of glucose from blood to astrocytes and then to neurons [86], the prolongation of synaptic activity at inhibitory synapses by diffusion of chloride across the astrocytic network [87], and the removal of excess neurotransmitters and ions (K^+^ and glutamate buffering) [88]. 

In addition, Cxs expressed by astrocytes enable communication between astrocytes and neurons by allowing the release of gliotransmitters. Gliotransmitters are neuroactive molecules that can lead to modification of synaptic activity and plasticity, and which in some cases are released by Cx hemichannels. As an example, astrocytes can release D-serine, a co-agonist of NMDAr, though the activation of Cx43 opening in response to high intracellular calcium. This was demonstrated by pharmacological and genetic inhibition of Cx43 that resulted in the reduction of the amplitude of postsynaptic excitatory currents mediated by NMDA receptors in the mouse prefrontal cortex [89].

Astrocytes also play a fundamental role in maintaining the BBB. Cx30 and Cx43 are widely expressed in astrocytic endfoot processes that contact cerebral vasculature. Consequently, they form part of the BBB and are essential for maintaining its integrity. Double knockout of Cx30 and Cx43 leads to edema of astrocyte endfeet, increased permeability of the BBB, and a loss of aquaporin 4 and β-dystroglycan. The latter is an important transmembrane receptor that anchors astrocytic endfeet to the perivascular basal lamina [90].

#### 3.1.3. Oligodendrocytes

Oligodendrocytes are the myelin-producing cells in the CNS. In addition to myelin synthesis, oligodendrocytes also have key functions such as metabolic supply and ion buffering, among others, [91] that are to some degree dependent on hemichannels and gap junctions. Oligodendrocytes derive from NPCs in the ventricular zone, specifically from oligodendrocyte precursor cells (OPCs) located in multiple regions of the developing CNS. Hemichannels are also expressed in OPCs and play a major role in the development of the oligodendroglia lineage. For instance, Niu et al., demonstrated that OPCs express Cx29 (Cx31.1 in humans) and Cx47, and that Cx-hemichannel mediated glucose uptake supports the proliferation of OPCs. They further demonstrated that this process was dependent on intracellular calcium to activate said Cx hemichannels [92]. Notably, Cx43 knockout in mice reduced glucose concentrations in the extracellular space, so much so that OPCs proliferation was reduced. Furthermore, Cx43 knockdown embryos have demonstrated a concomitant decrease in Sox10 positive cells [5], a key transcription factor in oligodendrocyte differentiation [93].

Cx hemichannels also play key roles in the normal function of the mature cell. Mature differentiated oligodendrocytes express Cx29, Cx32, Cx45 and Cx47, through which they communicate with other oligodendrocytes via homotypic gap junctions mainly formed by Cx47/Cx47 and by Cx32/Cx32 [94] or with astrocytes via heterotypic gap junctions through Cx47/Cx43. The latter is especially important to maintain CNS myelin and general homeostasis of the panglial syncytium by transporting excess K^+^ ions from the juxtaparanodal periaxonal space to blood vessels, and secreting nutrients including pro-myelinating factors [95,96]. In fact, in vitro and in vivo studies have demonstrated that depletion of either Cx43 or Cx47 affects the maintenance of white matter function and favors the expansion of demyelinated plaques [65,97]. Furthermore, myelin sheaths in the mature oligodendrocyte are also intraconnected by Cx32/Cx32 homotypic gap junctions in the paranodal region that enable metabolic transport, electrical coupling, nutrient transport and spatial buffering. Cx29 (Cx31.3 in humans) has also been detected at the internode of small myelinated fibers [98,99]. Consequently, Cxs and hemichannels play a key role in the development of oligodendrocytes, in the insurance of glial homeostasis, and in the maintenance of the CNS myelin. This is evidenced when mutations in these channels result in demyelinating disorders. For instance, mutations in the gene coding for Cx47 (*GJC2*) give rise to a hypomyelinating leukodystrophy. 

#### 3.1.4. Microglia

Microglia are the tissue-specific macrophages of the CNS first described by Pío del Rio Hortega during the 1920s. Since then, microglia have been broadly studied as to its functions and role in the CNS. Past decades have reinforced the theory which states microglia does not share a common progenitor with neurons and other CNS resident cells, instead, microglia derive from myeloid progenitors in the yolk sac that begin colonizing the CNS as early as embryonic day 9 [100,101]. Apart from their physiological functions as a phagocytic cell, they also provide support for neurons and have also been involved as regulators of brain development [102]. Interestingly, microglia express different genes during development and exert different functions, they can be divided into early microglia, pre-microglia and adult microglia depending on specific gene expression [103]. They mainly regulate brain development through the release of diffusible factors that support myelination, neurogenesis, axon fasciculation, synaptic formation and induction of cell death or survival. 

Whether the release of these diffusible factors is a Cx- or Panx-dependent process is not known. However, during the development of microglia, pannexons have been shown to regulate microglial morphology. Panx1-dependent release of ATP in response to glutaminergic signaling regulates dendritic morphology and the dynamic surveying movements in their processes [104]. The adult microglia express different Cxs and Panxs depending on their activation state. In the resting surveillance state Cx36, Cx32 and Panx1 are highly expressed, while in the activated states Cx expression increases, mainly involving Cx29, Cx32, Cx36, and Cx43. Inflammatory mediators such as TNF-a and INF-g can induce the opening and expression of Cx32, Cx43, and Panx1, which may result in excessive glutamate or ATP release causing excitotoxic neuronal death [105]. Thus, hemichannel-dependent glutamate release by microglia has acquired attention for the treatment of various neurodegenerative diseases. 

### 3.2. Role of Cxs and Panxs in Cells of the PNS

#### 3.2.1. Schwann Cells

Schwann cells are glia cells that surround the axons of the PNS, which can be classified as myelinating Schwann cells that will wrap around a single axonal segment (contrary to oligodendrocytes), and non-myelinating Schwann cells (also referred to as Remak Schwann cells) that aid various functions, mainly in the maintenance of the axon, peripheral nerve regeneration, neuronal survival and remodeling of the neuromuscular junction, where they are primarily located [106]. All Schwann cells derive from neural crest cells, a multipotent group of cells that are induced at the region where the edges of the folded neural plate meet. Neural crest cells will subsequently migrate throughout the embryo, where they will give rise to different types of cells, mostly contributing to the PNS, bones and cartilage of the cranio-facial area, heart, skin and some endocrine cells [107]. Once they approach peripheral nerves emanating from sensory and autonomic ganglia, they will further differentiate into Schwann cell precursors, then to immature Schwann cells, and after birth immature Schwann cells will differentiate into non-myelinating or myelinating Schwann cells [108,109].

Now, Cx43 appears to regulate some aspects of neural crest cell migration and development, which is after all essential for the formation of Schwann cells. Jourduil et al., demonstrated that Cx43 is expressed in premigratory and migratory cranial neural crest cells in chicks [5]. Interestingly, Cx43-depleted embryos exhibited two important characteristics: the first was a delayed emigration of neural crest cells from the neural tube and the second was having fewer premigratory cranial neural crest cells. This reduction was not because of apoptosis or reduction in cell proliferation, rather it was because of the role of Cx43 in the specification of premigratory cranial neural crest cells [5]. Notably, Cx43 knockdown embryos had a concomitant loss of Snail2 (Coded by *SNAI2* gene, Snail Family Transcriptional Repressor 2), Pax7 (Paired Box 7) and Sox10 (SRY-Box Transcription Factor 10), which are transcription factors that play a critical role during development [5]. Furthermore, Kotini et al., added that Cx43 can also act as a transcription regulator and could control neural cell crest migration by regulating the transcription of N-cadherin [56].

Adult Schwann cells also have functions that depend on Cxs and Panxs. Most importantly, Cx32 is involved in the maintenance of myelin, being expressed in the inner mesaxons, paranodal loops, Schmidt-Lanterman incisures, and in the two outer layers of myelin [110]. Cx32 plays such a critical role in the function of the Schwan cell that mutations in *GJB1* (the gene that codes for Cx32) lead to Charcot-Marie-Tooth disease. There are many animal studies describing the consequences of Cx32 knockout; however, the exact role of Cx32 in the PNS it is not clear from a molecular perspective. After electrical stimulation of the myelinated nerve, ATP is released, triggering a rise in cytosolic calcium and in the mitochondrial matrix of the Schwann cells surrounding the axon. Chronic suppression of this purinergic-mediated signaling has been shown in vivo to inhibit the correct formation of myelin and to cause hypomyelination [111]. 

Consequently, given that Cx32 hemichannels can release ATP, it is hypothesized that this calcium signaling could be amplified by Cx32, leading to ATP-induced ATP release, and therefore contributing to the correct formation and maintenance of myelin. Furthermore, Cx32 seems to play an important role for K^+^ buffering during neuronal activity. Increased periaxonal space and vacuolization is seen in Cx32 and Cx32/47 KO mice, respectively, and it is thought to be a consequence of periaxonal accumulation of K^+^ [112,113].

#### 3.2.2. Satellite Glial Cells (SGC)

SGC are flattened interconnected cells that surround neuronal cell bodies in ganglia of the PNS. They are found in sensory neurons from sympathetic and parasympathetic ganglia, and their close contact with neurons and the extracellular space determines their homeostatic function. They regulate neuronal microenvironment and respond to stressors activating an inflammatory profile, which is why they are also called the astrocytes of the PNS [114,115,116]. Interestingly, satellite glial cells share the same embryonic origin as neurons: they come from the neural crest and some evidence suggests they might also come from neuroepithelial cells that migrate from the spinal cord; however, their cellular lineage is still not clear [114,115,116].

Cx and Panx involvement in the PNS and gliotransmission is still yet to be fully elucidated; however, recent studies have not only shown the expression of connexons, pannexons, and hemichannels in sensory neurons and SGCs, but have also cast light on how its malfunctioning and dysregulation might lead to different pathologies such as chronic pain (in the case of SGCs) [117]. A study performed by Ohara et al., showed an increase in the expression of Cx43 in the trigeminal ganglion (TG) after chronic constriction injury (CCI) in a rat model, and congruently, reduction of Cx43 expression reduced pain in CCI rodents. However, when it came to uninjured rodents, knockdown of Cx43 resulted in pain-like behavior which stabilized once Cx43 levels normalized [118]. The aforementioned results evidence the complex neuroregulatory role SGCs have and suggest Cx43 might be involved in chronic pain but not in acute pain. Apropos Panx1, a study performed by Zhang et al., showed that Panx1 upregulation in the dorsal root ganglion (DRG) appeared 5 days after L5/L6 spinal nerve ligation (SNL) and lasted up to 3 weeks after the surgery [119]. Additionally, chronic orofacial pain models have consistently shown abolishment of tactile hypersensitivity after injection of Complete Freund’s Adjuvant (CFA) in the TG in Panx1-deficient mice [120].

Concomitantly, according to Kurisu et al., Panx1 inhibition and metabotropic glutamate receptor subtype 5 (mGluR5) antagonism have been shown to reduce orofacial neuropathic pain following infraorbital nerve injury in a rat model [121]. While previous results are supporting evidence to theories stating a relation between Cxs and Panxs in SGCs, further studies are still needed to completely understand the role these proteins play in chronic pain. 

## 4. Clinical Correlation

### 4.1. Congenital Neurological Diseases Associated with Cx and Panx Alterations

#### 4.1.1. Charcot-Marie-Tooth Disease

X-linked Charcot-Marie-Tooth disease (CMTX) is a hereditary neuropathy resulting from a mutation in the GJB1 (Gap Junction protein beta 1) gene responsible for the expression of Cx32 in Schwann cells and oligodendrocytes. Over 450 mutations of different types including missense, frame shift, deletion and non-sense have been related to this disease. Cx32 gap junction channels are indispensable for ionic passage, more specifically that of K^+^ and other signaling molecules across the myelin sheath in Schwann cells (Figure 4), while the lack or dysfunction of this Cx has been shown to cause this demyelinating neuropathy [122]. 

Additionally, mutations in Cx32 are likely to result in mitotic instability through calcium-calmodulin-dependent kinase type II overexpression, CAMKII plays an important role in oligodendrocyte maturation and CNS myelination [110]. Although many other tissues and cells, such as oligodendrocytes, express Cx32, peripheral neuropathy seems to be the sole manifestation of the GJ1B gene mutation, nevertheless case reports have described transient CNS manifestations which is why CMTX should be among the list of differential diagnoses in patients with peripheral neuropathy who present with sudden CNS impairment [126]. The mutations that lead to CMTX principally cause reduced functional coupling (assembly inhibition), reduction of pore size, increased sensitivity to acidification-induced closure, stabilization of the closed state of the channel, or inappropriate trafficking. Innovative treatment in this disease is mainly based on gene therapy, by GJB1 gene delivery [127,128].

#### 4.1.2. Oculodentodigital Dysplasia

Oculodentodigital dysplasia (ODDD) is a rare disease caused by a mutation in the gene GJA1 located on chromosome 6 that codes for Cx43. The most frequent inheritance pattern is autosomal dominant, but some cases of autosomal recessive inheritance have been reported [129]. It is characterized by craniofacial abnormalities (microcephaly, prominent columella, underdeveloped nasal alae) with various dental (hypoplastic enamel, small teeth, premature loss of teeth), limb (syndactyly, campodactyly), eye (microcornea, microphthalmia, alterations of the ocular adnexa) and skin alterations (palmoplantar keratoderma) [130]. Cardiac and neurological involvement have also been described from spasticity and hyperreflexia, white matter changes, calcification of the basal ganglia, gait disturbances, epilepsy, neurogenic bladder, and mental retardation to endocardial cushion defects, atrial and ventricular septum defects, among others [26,131,132]. Mutations associated with neurological manifestations have shown to diminish Cx43 gap junction intracellular communication, decrease the number of Cx43 gap junction plaques, increase hemichannel activity, or cause deletion of the first N-terminal aminoacids leading to alterations in Cx43 trafficking resulting in intracellular retention [133]. There is not a stepwise molecular explanation for the involvement of Cx43 and the neurological manifestations seen in ODDD, as most of the literature describes the role of Cx43 in the astrocyte and astrocytic network, and most of the physiopathological insights is based on Cx43 KO animal experiments. 

#### 4.1.3. Pelizaeus-Merzbacher-like Disease Type 1 (PMLD1)

PMLD1 is a slowly progressive leukodystrophy caused by mutations in the gene GJC2 that encodes Cx47. Cx47 is highly expressed in oligodendrocytes (Figure 4) and is involved in the formation and maintenance of myelin [134]. More than 50 mutations are associated with PMLD1, most of which are loss-of-function mutations. PMLD1 is characterized by nystagmus, ataxia, delayed acquisition of motor milestones, hypotonia, dysarthria, spasticity, extrapyramidal movement disorders and some patients may have mild intellectual disability [132]. Apart from the altered functions of Cx47 in oligodendrocyte gap junctions (nutrient transport, pro-myelinating factor secretion, K^+^ buffering), mutations in this protein may cause activation of the unfolded protein response, which could cause increased apoptosis [135]. In addition, proteolipid protein (PLP) is one of the main proteins in the CNS myelin sheath. It has been shown that the loss of Cx47 is associated with a concomitant loss of PLP [136] suggesting a role of Cx47 in supporting the integrity of myelin. Cx47 deficiency could also correlate with a reduction in UGT8A (UDP galactosyltransferase 8A) and FA2H (fatty acid 2-hydroxylase), which play an important role in myelin lipid metabolism. 

### 4.2. Acquired Neurological Diseases

#### 4.2.1. Multiple Sclerosis (MS)

Multiple sclerosis is a chronic immune-mediated inflammatory demyelinating disease of the CNS. The pathogenesis of MS is not clear; however, many pathways have been involved in tissue injury such as degeneration, inflammation, and demyelination. Immunopathogenesis of MS mainly involves imbalanced interactions between T cells, myeloid cells, B cells and regulatory cells [137]. Recently, B cells have demonstrated an essential role in the inflammatory and neurodegenerative components of the disease and new B-cell-directed therapies have been successful [138]. Antigen-triggered autoimmunity is another possible mechanism for the development of MS that develops secondary to molecular mimicry of a pathogen (principally believed to be Epstein-Barr virus) [139,140]. 

Oligodendrocyte apoptosis secondary to viral or glutamate induced excitotoxicity, chronic response to a viral infection, or a genetically determined neuroglial degenerative process have been some alternate theories for the development of this disease. After axonal injury secondary to immune-mediated mechanisms, remyelination becomes a crucial process to protect the axon from degeneration, and until MS can be prevented or treated before myelin is damaged, therapies targeting remyelination are needed [141]. Cxs support the proliferation of OPCs, mediate oligodendrocyte/oligodendrocyte and astrocyte/oligodendrocyte communication, and support nutrient transport to the myelin sheath. Crosstalk between oligodendrocytes and astrocytes (mediated by Cx47/43) is crucial for oligodendrocyte homeostasis and has been found to be relevant for both myelination and demyelination, principally in KO animal studies [65,97,141,142]. 

Alteration in Cx proteins have been found in MS patients and in MS animal models [143]. For instance, Markoullis et al., documented a decreased immunofluorescent staining of Cx32 and Cx47 in and around MS lesions of postmortem brain samples from MS patients compared to controls. Cx43, on the other hand, had an increased expression in the normal appearing white matter (NAWM). Furthermore, the authors found an increased expression of Cx47 in OPCs, but interestingly they showed a limited connection to astrocytes. These findings show that patients with MS have an alteration in the expression of Cxs that could contribute to the development of the disease [144] and that astrogliosis seen in MS could be responsible for the reduced connectivity between OPCs and astrocytes, thereby preventing their differentiation and remyelination. 

This hypothesis is possible given that astrocytes induce the proliferation of OPCs via Cx47-mediated activation of Chi3I1 (Chitinase 3-like protein 1) [145] a mechanism that could be altered in reactive astrocytes. In recent years, the same author found an increased expression of not only Cx43 but also Cx30 in cortical lesions that were accompanied by astrogliotic changes [144]. Astrocytes become reactive in response to the inflammatory environment seen in MS, and Cx43 gap junctions expressed in these astrocytes become activated hemichannels. The opening of hemichannels (especially Cx43) damages myelin via release of chemokines, further exacerbating inflammation [146,147] so much so that Cx43 deletion in astrocytes promotes CNS remyelination by modulation of local inflammation [148,149] in multiple sclerosis animal models.

Taking this into account, specific Cx43 hemichannel inhibitors would be useful in the pathophysiology of multiple sclerosis. Carbenoxolone, a gap junction blocker targeting Cx43, has been evaluated in animal models of MS. The authors found that carbenoxolone significantly reduced the severity of autoimmune encephalomyelitis (the most commonly applied model for MS) and reduced the quantity of inflammatory mediators (Table 3) [150]. More studies are needed to see if modulation of Cx43 hemichannel activation in MS could aid in the progression of the disease. To the best of our knowledge Gap19, a selective Cx43 hemichannel inhibitor, has not been tried in MS models.

#### 4.2.2. Neurodegenerative Diseases: Alzheimer’s Disease (AD), Parkinson’s Disease (PD)

AD is one of the most common forms of dementia and its prevalence is expected to increase in the following years. It is characterized by cognitive dysfunction and memory loss primarily due to cholinergic neuron loss, and posterior calcium and NMDA neurotoxicity. Major pathological characteristics in AD include neuronal degeneration, deposition of amyloid b (Ab) plaques and neurofibrillary tangles [159]. However, many pathways are involved in the onset and development of this disease, and recently the involvement of Cx and Panx proteins has sparked interest. Reactive gliosis is another major characteristic in AD, involving reactive astrocytes and activated microglia.

Reactive astrocytes undergo morphological and functional changes that include the opening of Cx43 hemichannels [123] and Panx proteins (Panx1) that could lead to excess ATP and glutamate release, causing neurotoxicity [160]. Activated microglia have also been shown to increase Cx expression upon exposure to neuroinflammation, and lead to the same release of neurotoxic compounds [105]. Increased immunoreactivity of Cx43 has been found around Ab plaques [161,162], and notably, Orellana et al., demonstrated that low concentrations of Ab_25-35_ increased Cx43 hemichannel activity not only in astrocytes, but also in microglia and in neurons [163]. Subsequently, Maulik et al., confirmed that in mouse primary astrocyte cultures treated with Ab_25–35_ there is an increase of Cx43 hemichannel activity, but there was also a reduced functionality of the gap junction assembly at the surface [152]. 

The underlying mechanism by which Ab plaques increase Cx43 hemichannel expression and decrease Cx43 gap junctions is unknown. Insights provided by the study of Maulik et al. [152] suggest that Ab causes retention of Cx43 in the ER/ERGIC (endoplasmic reticulum/Golgi intermediate compartment), explaining the decreased functional gap junction assembly. Impairment of gap junction function of Cx43 is especially relevant in the astrocytic network, as the latter has been involved in memory formation and cognitive function [164]. Knowing the role of Cxs and Panxs in AD, novel studies are emerging evaluating various Cx or Panx inhibitors.

Flores-Muñoz et al., documented that acute Panx1 blockade with probenecid reduced the activation of p38 mitogen-activated protein kinase (MAPK), a kinase that has been widely involved in early neurotoxic signaling (Table 3). Panx1 was also found to be overexpressed and overactive in mouse hippocampal tissue in neurons and reactive astrocytes near Ab. The authors also found an age-dependent overexpression of Panx1 that correlated with the amount of Ab deposition, suggesting an Ab-induced toxicity [75]. In addition, a specific hemichannel inhibitor called boldine, an alkaloid derived from the boldo tree, shown to inhibit astrocytic and microglia hemichannels but not gap junctions, was evaluated in a murine model of AD. The use of boldine in this model caused a reduction of ATP and glutamate release by astrocytes and activated microglia, alleviating hippocampal neuronal suffering [153].

The pathogenesis of PD has also been linked with alterations in Cx and Panx expression. This disease is characterized by the loss of dopaminergic neurons in the striatum and substantia nigra, aggregations of α-synuclein (Lewy bodies), and astrogliosis in the substantia nigra. Common animal models of PD have shown increased expression of Cx30 and Cx43 in the striatum, and increased Cx43 hemichannel activity. In a murine model of PD, increased Cx43 hemichannel activity led to increased intracellular calcium in astrocytes and increased neuronal loss, and interestingly the administration of gap19, a Cx43 blocker, reverted dopaminergic neuron loss and microglia activation [155].

Fujita et al., provided novel insights into the role of Cxs in PD, suggesting Cx30 might have a role in astrocyte neuroprotection. This was demonstrated in a PD murine model with Cx30 knockout mice where, compared to wild-type mice, neuron loss was accelerated [165]. Furthermore, it has been demonstrated that a-synuclein might affect astrocytic hemichannel function, enhancing the opening of Cx43 and Panx1 in mouse cortical astrocytes. This abnormal opening generates alterations in intracellular calcium levels, gliotransmitter release, cytokine release, activation of nitric oxide synthase, and astrocyte survival [154].

#### 4.2.3. Pain

Panxs and Cxs have also been associated with the physiopathology of some types of pain, mainly neuropathic pain. This association has been principally related to the ability of Panx1 and Cx43 hemichannels to release ATP with subsequent activation of purinergic signaling. Purinergic receptor activation (P2X3, P2X4, P2Y) has been linked with pain signaling, thus Cx and Panx have been evaluated in several pain models as potential therapeutic targets [166]. For instance, it is known that vincristine, a commonly used chemotherapeutic drug, causes dose-dependent peripheral neurotoxicity that manifests as neuropathic pain. Li et al., demonstrated in a murine model that vincristine induced-neuropathic pain is accompanied by overexpression of Cx43. KO or inhibition of Cx43 by gap27 (a selective gap junction blocker) relieved pain hypersensitivity and reduced the release of inflammatory factors [156]. 

Furthermore, suppression of Cx43 expression in rats was shown to attenuate mechanical hypersensitivity after L5 spinal nerve ligation. Western blot analysis showed that the reduction in pain correlated with downregulation of Cx43 expression [167]. Many studies have also detected overexpression of Cx43 in pain models, supporting the hypothesis that Cx43-mediated ATP release induces pain [168,169]. However, a recent study performed by Zhang et al., showed contradictory results, suggesting that the antinociceptive properties of spinal phosphodiesterase-4 inhibitors (rolipram and roflumilast) in neuropathic pain were due to an increase in Cx43 expression in spinal astrocytes [170]. Furthermore, Panx1 has also been shown to be upregulated in various murine pain models, with effective reduction in pain measurements (mechanical allodynia, and behavioral responses) when probenecid (a Panx1 inhibitor) was administered [171,172,173].

#### 4.2.4. Epilepsy

Epileptic activity is known to be a result of hypersynchronous electrical stimuli as well as a disbalance between excitatory and inhibitory control mechanisms. While its exact pathogenesis comprises multiple mechanisms, astrocytic dysfunction has been broadly studied as a perpetuator and crucial actor. As previously stated, Cx43 and Cx30 enable astrocyte communication and the formation of the astrocytic network, which is responsible for the redistribution of ions, neurotransmitters, and metabolites along the astrocytic syncytium. When astrocytes become uncoupled due to gap junction or Cx abnormalities, extracellular K^+^ concentrations cannot be buffered, which might contribute to epileptiform activity and neuronal hyperexcitability [174]. Deshpande et al., demonstrated that astroglial Cx-deficient mice had higher seizure and interictal spike activity, implying that loss of gap junction coupling between astrocytes promotes neuronal hyperexcitability [175]. 

Furthermore, gap junction dissociation may result in increased hemichannel activity, which may enhance epileptiform activity via Ca^2+^ wave propagation and the release of ATP and glutamate [176]. There have been few studies examining the role of Cx43 hemichannel activity in epilepsy, owing to the difficulty of selectively blocking gap junction function but not hemichannel function. Nonetheless, some studies have discovered elevated Cx43 levels at the mRNA or protein levels. For example, immunohistochemistry analysis of brain epileptic foci extracted from patients with refractory epilepsy who underwent surgical treatment revealed increased expression of Cx43 and Cx32 compared to the control group [177].

Another study performed by Vincze et al., used an antibody against the gating peptide segment of Cx43 which specifically blocked Cx43. This blockage resulted in complete abolition of seizure-like events in five out of five hippocampal-entorhinal slices from rats; nevertheless, authors reported observing a residual non-epileptiform activity [178]. As for Cx36, divergent evidence has hindered the possibility of reaching a consensus on whether said Cx has an anti-epileptic or pro-epileptic effect. A study performed by Jacobson et al., concluded that Cx36 knockout mice have a higher susceptibility to pentylenetetrazol-induced seizures, thus suggesting Cx36 acts as an anticonvulsant [179]. Endorsing the anti-epileptic theory, a recent study performed by Brunal et al., sought to understand the reciprocal relation between Cx36 and neuronal hyperactivity using the zebrafish pentylenetetrazol-induced seizure model, finding evidence suggesting that Cx36 prevents hyperactivity within the brain and that the loss of Cx36 increases susceptibility to hyperactivity [180,181]. On the other hand, Gajda et al., reported how the administration of Quinine, which acts as a Cx36 blocker, led to a decrease in seizure activity. Lastly Vincze et al., observed a persistence in seizure activity even after administration of quinine in at least 60% of the rat hippocampal-entorhinal slices [178]. Considering the contrasting evidence, further studies are still needed to make definitive conclusions.

Panx1 and Panx2 channel overexpression has also been linked to epilepsy. When Li et al., examined surgical tissue from patients with focal cortical dysplasia, a common cause of childhood epilepsy, they discovered increased expression of Panx1 mRNA and protein in all samples, and of Panx2 only in certain subtypes of lesions. Notably, the level of Panx1 expression positively correlated with the frequency of seizures [76]. Consistently, studies have found overexpression of Panx1 in patients presenting with epileptic syndromes when compared to control patients [182].

Modulation of Panx1, either pharmacologically with probenecid or via genetic deletion, was evaluated by Aquilino et al., who found that seizure activity was suppressed by both methods across all epilepsy models in mice (injection of pentylenetetrazol, acute electrical kindling of the hippocampus and exposure to 4-aminopyridine) [157]. Panx1 is believed to sustain seizures through the release of ATP that can activate purinergic receptors, including P2X7. This receptor has been found to modulate neuronal excitability, contribute to microglia activation, and in various neuroinflammatory responses [183]. These receptors are also upregulated in different neuropathological conditions including Temporal Lobe Epilepsy and Rasmussen Encephalitis, two chronic disorders characterized by refractory seizures and persistent neuroinflammation [184].

#### 4.2.5. Autism Spectrum Disorder (ASD)

ASD is a group of neurological and developmental diseases characterized by sensory disturbances, repetitive behaviors, impaired social communication, and impaired adaptive functioning. The etiology is thought to be multifactorial, with both genetic and non-genetic factors contributing to its pathogenesis [185]. Given the multiple functions of Cxs in the homeostasis of brain tissue, the role of Cxs in the development of ASD has been evaluated. Fatemi et al., demonstrated that Cx43 expression is increased in Brodmann area 9 in subjects with autism [158], while activation of P2X2 receptors in the prefrontal cortex by astrocyte-derived ATP has been shown to modulate ASD-like behavior in mice. Whether this astrocyte release of ATP is mediated by Cx hemichannels has not been determined but given the pathogenic role of Cx43 hemichannels in other neurological diseases, it could be a possible explanation [186]. 

Furthermore, thanks to research in recent years it is now believed that gut microbiota plays a role in the pathogenesis of ASD through the gut-brain axis. Immune- and inflammation-mediated pathways may be the mechanisms by which gut microbiota contributes to the development of ASD. Wang et al., found that Cx43-related gut-derived epitopes were upregulated in the stool samples of ASD patients compared to controls, suggesting it may interfere with the normal function of human Cx43 [187].

#### 4.2.6. Cerebral Ischemia

After cerebral ischemia, hemichannels are unusually opened, allowing the entry of Na^+^ and Ca^2+^ and the release of ATP that ultimately leads to osmotic imbalance, energy depletion, and cell death [188]. ATP release from hemichannels can also activate microglia and increase the release of inflammatory factors that can further activate Cx43, creating a toxic cycle [189]. Astrocytic Cx43 therefore plays an important role in CNS ischemic injury. Based on this idea, Chen et al., evaluated the use of gap19, a selective Cx43-hemichannel inhibitor, on mice after induced middle cerebral artery occlusion. The results showed improvement in neurological scores, infarct volume reduction, attenuated white matter damage and reduced expression of inflammatory markers. Notably, gap19 inhibited the activation of the TLR4 (Toll-like receptor 4) pathway, a well-known mechanism for secondary injury after cerebral ischemia [39]. 

Panx1 channels have also been implicated in cerebral ischemic injury and inflammatory response. Wei et al., evaluated the role of a Panx1 inhibitor (^10^Panx) on rats after middle cerebral artery occlusion. ^10^Panx-treated rats showed reduced infarct volume, alleviation of neurological deficit, and reduced post-ischemic neuronal death. This effect was accomplished in part through a reduction of microglial activation and by inhibition of RIP3 (Receptor-interacting protein kinase-3 expression), a key protein involved in necroptosis (programmed necrosis) [190].

#### 4.2.7. Neuroinflammation

Microglia and astrocytes play a central role in neuroinflammation as they both have the potential of promoting inflammatory cascades [191]. Cx43 and Panx1, are central molecular structures in these cells, and different studies have highlighted them as key in inflammation pathways. Research on the neuroinflammatory profile of astroglia hemichannels when exposed to alcohol showed that ethanol triggers a proinflammatory state via an increase in Cx43 and Panx1 activity which favors the release of both ATP and glutamate, stimulating inflammatory cascades with microglial activation. This positively correlated with increased levels of interleukin 6 (IL-6), tumor necrosis factor alfa (TNF-α) and interleukin 1-beta (IL-1β); there was also a positive feedback loop for a proinflammatory state as cytokines stimulated p38 mitogen-activated protein kinases (p38 MAPK) which led to Cx and Panx opening, once again allowing for an increase in ATP levels and microglia activation [191].

This inflammatory component was also seen in primary cultured astrocytes subjected to oxygen deprivation/reperfusion injury, whereby all in all, hemichannel activation increased ATP release, which increased microglia activation and cytokine production of TNF-a and IL-1b [192]. Another study carried out by Xinyu Wang et al., demonstrated that Cx43 degradation prevented cell apoptosis and reduced TNF-a during ischemia, which adds evidence to the key role of this hemichannel during neuroinflammation [193]. All the research presented supports placing Cx and Panx in the center of the discussion about neuroinflammation and provides evidence to encourage the study of new pharmacological agents that aim specifically at these cellular structures when treating inflammationHigh yield treatments for neuroinflammation are raising in all areas of study, as seen in new treatments for neuroinflammation caused by irradiation (prevalent in cancer patients), innovative gene therapy may be upcoming in the following years as, Wei Zegn. Et al research with in vitro cells showed that microRNA-206 can relieve irradiation induced neuroinflammation, this was the case of transfected cells with miR-206 that not only reduced the inflammatory profile but also reduced Cx43 expression despite of being exposed to radiation [194]. 

## 5. Conclusions

Cxs and Panxs are widely expressed in the human body, and are essential for several functions in all organ systems. They regulate important aspects of differentiation, migration, specification, and proliferation during the development of cells in the CNS and the PNS, but they also have major functions in the adult CNS and PNS. They mainly exert these functions by modulation of purinergic receptor signaling and intracellular communication. Specific Cx isoforms have been related to specific diseases of the CNS and the PNS. Most of the information we have about Cx and Panx function is derived from animal studies in which mutations lead to a certain defect and thus their function is assumed. However, the role of Cxs and Panxs in diseases of the CNS and PNS is not clearly elucidated in a stepwise molecular manner. In neurodegenerative diseases, Cxs and Panxs are mainly involved due to overactivation of astroglial Cxs in response to the morphological and functional changes in reactive astrocytes. Overexpression of activated hemichannels results in the release of substances such as ATP or glutamate that led to neurotoxicity. Cx43 and Panx1 were the most common proteins associated with disease. Studies evaluating Cx43 and Panx1 inhibition by different compounds have yielded promising results in neurodegenerative diseases, stroke, pain and epilepsy. However, more studies are needed to corroborate these results and move forward with these exciting therapies. 

Cxs and Panxs are important players in the pathogenesis of many diseases. Preclinical studies have shown potential protective effects of Cx or Panx inhibition in diseases such as AD, pain and cerebral ischemia, among others. Thus, their role as potential therapeutics has been considered. There are many available agents that inhibit Cx and Panx channels such as chemical- (carbenoxolone), peptide- (gap19) or antibody-based inhibitors that could alter disease pathogenesis and provide a clinical benefit. Inhibiting the activity of Cx and Panx channels is challenging because it could interfere with physiological functions dependent on gap junctions, and obtaining selectivity for a specific Cx hemichannel can be difficult. Clinical trials have been held using activators of Cxs and Panx channels such as rotigaptide for treating atrial fibrillation (NCT00137332) and endothelial dysfunction (NCT00901563), and Cx43 inhibitors for the treatment of severe ocular surface burns. However, to the best of our knowledge there are no clinical trials evaluating Cx or Panx inhibitors for the treatment of neurological diseases.

## Figures and Tables

**Figure 1 biomedicines-10-02237-f001:**
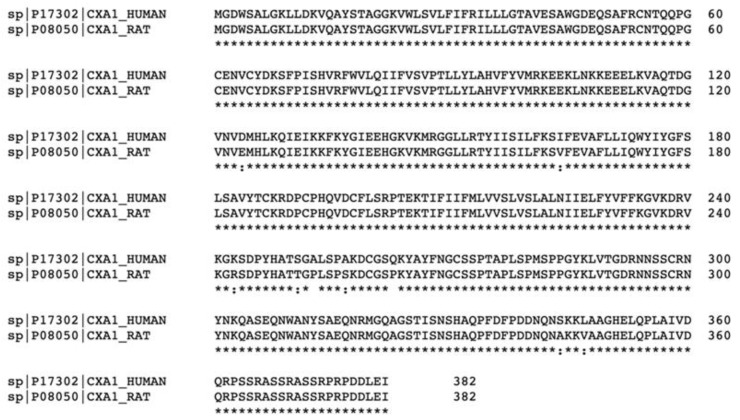
Sequence alignment of human Cx43 (also known as CXA1) and rodent Cx43 showing 97.64% sequence homology. Created with Uniprot.org, accessed on 17 August 2022. An * (asterisk) indicates positions which have a single, fully conserved residue. A : (colon) indicates conservation between groups of strongly similar properties. A . (period) indicates conservation between groups of weakly similar properties.

**Figure 2 biomedicines-10-02237-f002:**
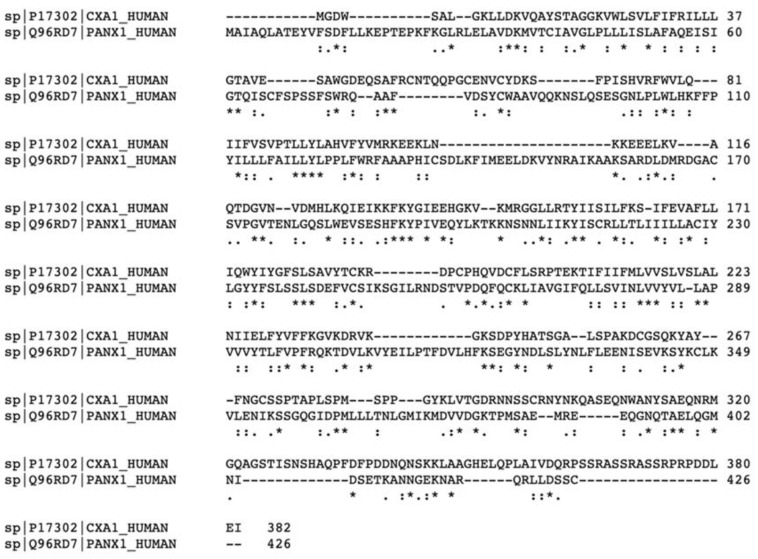
Sequence alignment of human Cx43 (also known as CXA1) and Panx1, showing 23.01% homology in their sequences. Created with Uniprot.org, accessed on 17 August 2022. An * (asterisk) indicates positions which have a single, fully conserved residue. A : (colon) indicates conservation between groups of strongly similar properties. A . (period) indicates conservation between groups of weakly similar properties.

**Figure 3 biomedicines-10-02237-f003:**
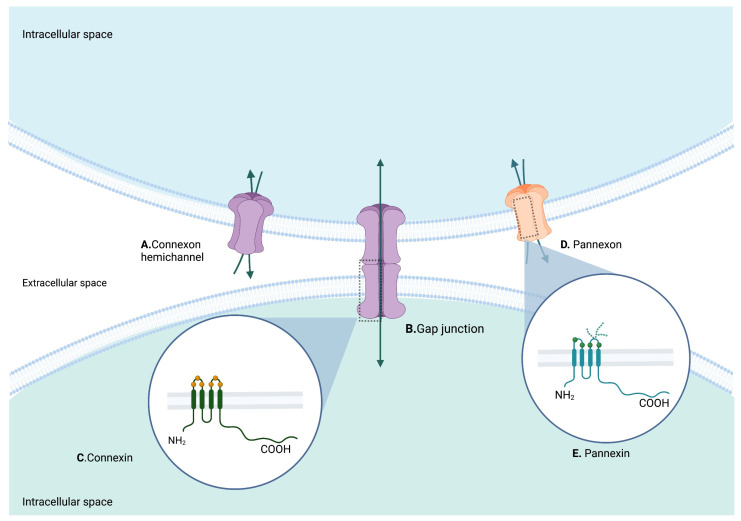
Structure representation of Cxs, Panxs, connexons, pannexons and gap junctions. (**A**) Oligomerization into hexamers of Cxs forming a connexon, or hemichannel, that allows the flow of small molecules. (**B**) Two connexons joined together forming a gap junction, allowing communication between two cells (**C**) Cxs have 4 transmembrane domains, cytosolic N-(amino) and C-(carboxyl) termini, two extracellular loops and one intracellular loop. They have 3 cysteine residues in each extracellular loop. (**D**) Pannexons formed by Panx1 have recently been found to oligomerize into heptameres. (**E**) Panxs consist of 4 transmembrane domains, with 2 extracellular loops, a cytoplasmatic loop, and cytosolic N-and C-termini. Panxs have 4 conserved cysteines—two in each extracellular loop—and they exhibit glycosylation of the first extracellular loop in Panx2 (not shown) and the second in Panx1 (shown). Created with BioRender.com. Accessed on 1 August 2022.

**Figure 4 biomedicines-10-02237-f004:**
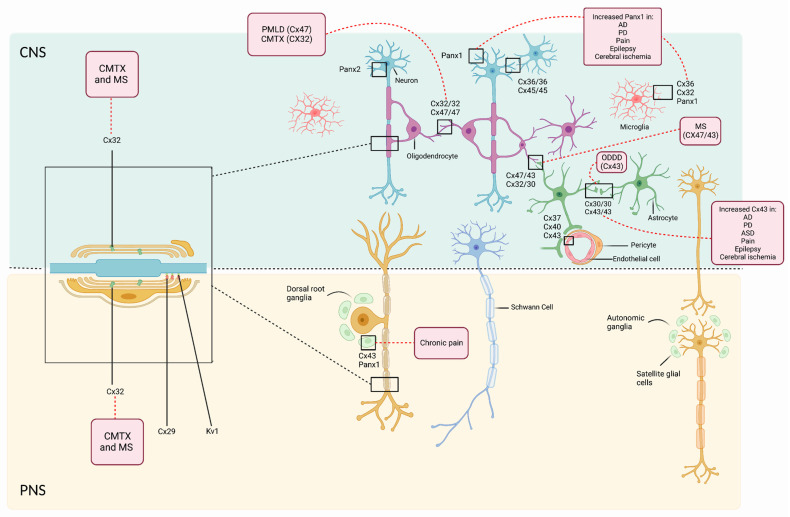
Expression of Cxs and Panxs in the CNS and PNS, and related diseases. In the upper figure the distribution of Cxs and Panxs can be seen along the CNS. Oligodendrocytes communicate with other oligodendrocytes via homotypic gap junctions mainly formed by Cx47/Cx47 and by Cx32/Cx32 [94]. Cx43 and Cx30 are expressed in astrocytes, and they allow the formation of highly interconnected astrocytes that give rise to the astroglial network; they are also expressed in astrocytic endfoot processes that contact cerebral vasculature. Astrocytes communicate with other astrocytes via homotypic gap junctions formed by Cx30/Cx30 or by Cx43/Cx43. With astrogliosis, astrocytes undergo morphological and functional changes that include the increased activation of Cx43 hemichannels involved with the release of neurotoxic compounds. This alteration has been related with AD, PD, ASD, Pain, Epilepsy, and Cerebral Ischemia [123]. Activated microglia have also been related with the release of neurotoxic compounds, and they express Cx36, Cx32, and Panx1 that can be induced by TNF-a and INF-b [105,124]. Inherited diseases have been related to mutations in the Cx genes: ODDD is caused by mutations in GJA1 gene that codes for Cx43 [125] mutations in the GJB1 gene that codes for Cx 32 expressed in Schwann Cells (shown in the lower part of the figure, PNS) and oligodendrocytes results in X-linked Charcot-Marie-Tooth Disease; PMLD is caused by mutations in the GJC2 that codes for Cx47. AD: Alzheimer’s disease, PD: Parkinson’s disease, CMTX: X-linked Charcot-Marie-Tooth disease, MS: multiple sclerosis, PMLD: Pelizaeus–Merzbacher-like disease type 1, ODDD: Oculodentodigital dysplasia, CNS: central nervous system, PNS: peripheral nervous system, ASD: autism spectrum disorder, GJA1: gap junction protein alpha 1, GJB1: gap junction protein beta 1, GJC2: gap junction protein gamma 2. Created with BioRe.nder.com.

**Table 2 biomedicines-10-02237-t002:** Functions of Panxs during development, in the adult brain, and related neurological diseases.

Panx	Gene	Relevant Function during Development	Relevant Function in the Adult Brain	Related Diseases and Functional Alteration	Reference
Panx1	*PANX1*	◦ Promotes the growth and proliferation of ventricular zone neural precursor cells in vitro and in vivo	◦ Modulates the induction of excitatory synaptic plasticity ◦ Contributes to neuronal death under inflammatory conditions ◦ ATP-dependent ATP release ◦ Increases expression with age	◦ Epilepsy ◦ Alzheimer’s disease	[19,30,75,76]
Panx2	*PANX2*	-	-	◦ Epilepsy	[76]

**Table 3 biomedicines-10-02237-t003:** Acquired diseases associated with Cx and Panx expression, related mechanisms for disease, and available studies on pharmacological modulation.

Disease	Cx or Panx Involved	Mechanism for Disease	Channel Modulation Studies	Reference
**MS**	Cx47, Cx43, Cx30, Cx32	◦ Reactive astrocytes have ↑ Cx43 (increased expression is correlated with inflammatory load) and ↑ Cx30 ◦ ↓ Cx47 ◦ ↓ Cx32	◦ **Carbenoxolone** (Cx43 blocker) reduced severity of autoimmune encephalomyelitis (a MS animal model)	[144,150]
**AD**	Cx43, Panx1	◦ ↑Cx43 hemichannel activity ◦ ↓ Cx43 gap junction assembly ◦ ↑ Panx1 activity (leads to early synaptic dysfunction)	◦ **Probenecid** (Panx1 blocker)—mitigates early synaptic plasticity defects (mouse model) ◦ **Boldine** (Cx43 hemichannel inhibitor in astrocytes and microglia)—↓ ATP and glutamate release by astrocytes and activated microglia, alleviated hippocampal neuronal suffering (mouse model)	[75,151,152,153]
**PD**	Cx43, Panx1	◦ α-synuclein-induced ◦ ↑ Cx43 hemichannel activity	◦ **Gap19**, (Cx43 blocker)—reverted dopaminergic neuron loss and microglia activation in a murine model of PD	[154,155]
**Pain**	Cx43, Panx1	◦ ↑ Cx43 hemichannel activity (Release of ATP and chemokines contributes to neuropathic pain) ◦ ↑ Panx1 activity	◦ **Gap27** (selective hemichannel inhibitor, Cx43 mimetic peptide)—Reduced pain hypersensitivity and the release of inflammatory factors	[156]
**Epilepsy**	Cx43, Cx30, Panx1, Panx2	◦ ↑ Cx43 hemichannel activity ◦ ↓ Astroglial gap junctions ◦ ↑ Panx1 and Panx2 expression	◦ **Probenecid** (Panx1 blocker)—suppressed seizure activity in epilepsy models in mice	[157]
**ASD**	Cx43	◦ ↑ Cx43 expression	-	[158]
**Cerebral ischemia**	Cx43, Panx1	◦ ↑ Cx43 hemichannel activity	◦ **Gap19**—neuroprotective effects on cerebral ischemia/reperfusion, infarct volume reduction in mice after middle cerebral artery occlusion, decreased Cx43 expression and inflammatory cytokines	[39]

PLP: proteolipid protein, UGT8A: UDP galactosyltransferase 8A, Fa2h: (fatty acid 2-hydroxylase).

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
