# Peer review of "Connexins and Pannexins: Important Players in Neurodevelopment, Neurological Diseases, and Potential Therapeutics"

_biomedicines, 2022, doi:10.3390/biomedicines10092237_

Round 1

Reviewer 1 Report

The manuscript by Daniela Bracaldo-Santamaria et al.  reviewed the recent molecular and structural studies of connexins and pannexins. The current manuscript is on a topic of relevance to the readers of the journal, especially for the “Connexins and Pannexins in Embryonic and Fetal Development” special issue. I found the topic of this paper is interesting and has listed lots of information. However, a summary of all those researches with insights for the future direction is not there. Please see comments below.

Comments:

1. For Title, “Connexin and pannexin role’ should be ‘The role of connexins and pannexins’. Also, it’s not common to use a question as the title. Some modifications are needed here.

2.The coherence in abstract paragraph can be improved. For example, in Line 17, split this sentence at “recent”. Link citations here for the recent studies. Move the sentences from “Cell-to-cell communication via gap junctions” to “Consequently,…” right after that sentence.

3. In the introduction part, add more citations. Every fact needs to have a reference here.

4.Line 19, ”although” is not used correctly (also incorrect at line 24&82). Maybe change this sentence to something like “Despite…, …” or “…;however,  …”;The sentence from line 42-44 is confusing. Please slit it up or add “,” before “juxtracrine”. Also, “cell-to-cell communication is essential” appeared at line 40, please rephrase it.

5.In the introduction, the author mentioned there are 21 Cxs in rodent and 20 in human. Maybe a figure here with their molecular weight and sequence alignment can be useful for the audience.

6.In the introduction line 75, innexins were mentioned which might not be necessary. Also, it might be helpful to move figure 1 here to give the audience a better impression of the structure similarity.

7. The cartoons in Figure1 are not “structure”, more or less “topology” or “cartoon representation”. Also, Figure 1c is incorrect. First, the Connexon channels should span across two layers of lipid bilayers to form the gap junction. In that case, extracellular space should be sandwiched between lipid bilayers, and cytosol spaces are on both top of bottom of the figure. Also, the subunits should be more or less identical with size and the number should agree with reality. Now the blue channel is a tetramer, which is not right. Note that lots of high-resolution (beyond 4 Angstrom) structures of connexin and pannexin channels have already been solved. Maybe use those instead or change the figure.

8. A sequence alignment here for Panxs and Cs might be helpful at line82.

9.Please add citations for at 83 for the debate whether Panxs are able to form gap junction channels with evidence from both sides.

10.Line106-108 should be separated from the rest of paragraph where the author talked about the importance of this manuscript.

11. Please add a paragraph reviewing all the structures available for connexin, pannexin (maybe innexin if the author decided to mention it) part2 before jumping to Cx43. Maybe mention the resolutions, species (goat, human, mice?), sequence information (any mutation/truncation?) and methods (x-ray, cryoEM, NMR?) for those structures.

12. Part3&4 can be restructured together and condensed.

13.Is line 505-526 a figure description? Shouldn’t it be put in figure caption? Instead of just a description of figure, a summary of all the recent clinical research is needed here before jumping to specific diseases.

14. The conclusions did not give insights on whether connexins and pannexins are potential therapeutic targets. Are there any clinical trials? Which phase are they? Either add more vistas for this paragraph or change the title to something less focusing on therapeutic potentials.

Author Response

Dear reviewer, thank you for taking the time to review our manuscript. 

We have attached a word document with our responses.

Reviewer 2 Report

The Ms requires extensive editing concerning English grammar and style. In addition, Authors are invited to refresh discussion and references concerning the role for intercellular connexin channels in Epilepsy.

Author Response

Dear reviewer, 

Thank you for taking the time to review our manuscript. 

Here are the answers to your comments:

The Ms requires extensive editing concerning English grammar and style. In addition, Authors are invited to refresh discussion and references concerning the role for intercellular connexin channels in Epilepsy.

-We hired a native English speaker to help with the grammar and style, and we refreshed the discussion and references in epilepsy.

We attached a PDF with a certification of proofreading by a native english speaker. 

Reviewer 3 Report

In this manuscript, Baracaldo-Santamaria et al. provide a comprehensive review of the pathophysiological roles of connexins and pannexins in the nervous system. The paper includes a general introduction of connexins and pannexins, followed by their function and expression in the neurological system and their involvement in the initiation and development of various nervous diseases. The paper is well organized and, in most parts, well written. The tables are well-made and helpful in understanding the content. 

I only have several general comments for improvement.

1) The manuscript needs to be polished in English. Grammatical errors, including those in the tables, need to be corrected.

2) Paragraphs are too lengthy. It is hard to get the points. Efforts should be made to increase the readability. 

3) An additional short paragraph should be added, addressing whether connexins or pannexins could be potential therapeutic targets, as mentioned in the title. The advantages, disadvantages and challenges of the therapies should also be briefly described.  

Author Response

Dear reviewer, 

Thank you for taking the time to review our manuscript. 

We attached a word document with the answers to your comments. 

Round 2

Reviewer 1 Report

Thank you for addressing my comments. I am satisfied with the revision.